# EinSteinVI: General and Integrated Stein Variational Inference

## Abstract

Stein variational inference is a technique for approximate Bayesian inference that has recently gained popularity because it combines the scalability of variational inference (VI) with the flexibility of non-parametric inference methods. While there has been considerable progress in developing algorithms for Stein VI, integration in existing probabilistic programming languages (PPLs) with an easy-to-use interface is currently lacking. EinSteinVI is a lightweight composable library that integrates the latest Stein VI methods with the PPL NumPyro (Phan et al., 2019). EinSteinVI also provides our novel algorithm ELBO-within-Stein to support the use of custom inference programs (guides), in addition to implementations of a wide range of kernels, non-linear scaling of the repulsion force (Wang & Liu, 2019b) and second-order gradient updates using matrix-valued kernels (Wang et al., 2019b). We illustrate EinSteinVI using toy examples and show results on par with or better than existing state-of-the-art methods for real-world problems. These include Bayesian neural networks for regression and a Stein-mixture deep Markov model, which also shows EinSteinVI scales to large models with more than 125,000 parameters.

## 1 Introduction

Interest in Bayesian deep learning has surged due to the need for quantifying the uncertainty of predictions obtained from machine learning algorithms (Wilson & Izmailov, 2020; Wilson, 2020). The idea behind Bayesian inference is to describe observed data $\mathbf{x}$ using a model with latent variables $\mathbf{z}$. The goal is to infer a posterior distribution $p(\mathbf{z}|\mathbf{x})$ over the latent variables given a model describing the joint distribution $p(\mathbf{z}, \mathbf{x}) = p(\mathbf{x}|\mathbf{z})p(\mathbf{z})$. We obtain the posterior by following the rules of Bayesian inference:

$$p(\mathbf{z}|\mathbf{x}) = Z^{-1}p(\mathbf{x}|\mathbf{z})p(\mathbf{z})$$

where $Z = \int_{\mathbf{z}} p(\mathbf{x}|\mathbf{z})p(\mathbf{z})\mathrm{d}\mathbf{z}$ is the normalization constant. For most practical models, the normalization constant lacks an analytic solution or requires an infeasible number of computations, complicating the Bayesian inference problem.

Variational Inference (VI) techniques (Blei et al., 2017; Hoffman et al., 2013; Ranganath et al., 2014) aim to approximate the posterior distribution. VI poses a family of distributions over the latent variables $q(\mathbf{z}) \in \mathcal{Q}$ and finds

$$\arg\min_{q \in \mathcal{Q}} D(q(\mathbf{z})|p(\mathbf{z}|\mathbf{x})),$$

where $D$ is a divergence[1] between the variational distribution $q$, also called the guide, and the true posterior distribution $p(\mathbf{z}|\mathbf{x})$. The Kullback-Leibler divergence is a typical choice. VI often provides good approximations that capture uncertainty and scales to millions of data points by a suitable choice of $\mathcal{Q}$ and inference method.

Stein VI is a family of VI techniques for approximate Bayesian inference based on Stein's method (see Anastasiou et al. (2021) for an overview) that is gaining popularity since it combines the scalability of traditional VI with the flexibility of non-parametric particle-based methods. Stein variational gradient descent (SVGD) (Liu & Wang, 2016) is a recent Stein VI technique which uses a set of particles $\{\mathbf{z}_i\}_{i=1}^N$ as the approximating distribution $q(\mathbf{z})$. As a particle-based method, SVGD is well suited for capturing correlations between latent variables. The technique preserves the scalability of traditional

---

[1]An asymmetric distance that might satisfy the triangle inequality.

VI approaches while offering the flexibility and modeling scope of techniques such as Markov chain Monte Carlo (MCMC). SVGD is good at capturing multi-modality (Liu & Wang, 2016; Wang & Liu, 2019a), and has useful theoretical interpretations such as a set of particles following a gradient flow (Liu, 2017) or in terms of the properties of kernels (Liu & Wang, 2018).

Many advanced inference methods based on SVGD have been recently developed; these include non-linear Stein (Wang & Liu, 2019a), factorized graphical models (Zhuo et al., 2018; Wang et al., 2018a), matrix-valued kernels (Wang et al., 2019a) and support for higher-order gradient-based optimization (Detommaso et al., 2018). These techniques have extended the scope of SVGD, allowing more flexible and accurate approximations of the posterior distribution. While algorithmic power is growing, a distinct lack of integration with a general probabilistic programming language (PPL) framework remains. Such integration would solve one of the most prominent limitations of PPLs with traditional VI: their lack of flexibility in capturing rich correlations in the approximated posterior.

The main problem with SVGD is that it suffers from the curse of dimensionality: the number of particles required to adequately represent a posterior distribution is exponential in its dimensionality. Nalisnick & Smyth (2017) suggest resolving this by using Stein mixtures and propose an inference algorithm that uses differentiable non-centered parameterization (DNCP) (Kingma & Welling, 2014) and importance weighted Monte Carlo gradients (Burda et al., 2015).

In this article we propose a novel algorithm for inference of Stein mixtures called ELBO-within-Stein. Unlike prior work on Stein mixtures (Nalisnick & Smyth, 2017), ELBO-within-Stein i) is simple to implement, ii) only has the learning rate as a tuneable hyper-parameters, iii) requires only one gradient evaluation of the ELBO for each particle (10 is typical for Stein mixtures) and iv) supports the use of a tailored guide. We validate our algorithm experimentally, showing significant improvements in accuracy. ELBO-within-Stein is the core algorithm of our Stein VI library called EinSteinVI library. Our library further includes SVGD as a special case of ELBO-within-Stein and all the advanced methods mentioned above. The only other PPL with Stein VI methods is `PyMC3` (Salvatier et al., 2016). However, `PyMC3` does not include any of the advanced Stein VI methods mentioned above or Stein mixture methods.

The EinSteinVI library extends the NumPyro PPL (Bingham et al., 2019; Phan et al., 2019). NumPyro is a universal probabilistic programming language (van de Meent et al., 2018) for Python, which allows arbitrary code to be executed in both its model and guide. The computational backend of NumPyro is `Jax` (Frostig et al., 2018), which gives access to the powerful program optimization and parallelizability provided by the `Jax` compiler. As EinSteinVI works with arbitrary guides, NumPyro is a well suited language for embedding EinSteinVI. This is because NumPyro i) is embedded in Python, the de facto programming language for data science, ii) includes the necessary data-structures for tracking random variables in both model and guide, iii) features SVI with an API that is highly suitable for EinSteinVI, and iv) benefits computationally from Jax. Our extensions include SVGD, Stein-mixtures formulated as ELBO-within-Stein, and the advanced methods mentioned above.

Concretely, our contributions are:

- **ELBO-within-Stein** (EinSteinVI). A novel algorithm for Stein mixtures that only requires a single gradient evaluation of the ELBO per particle.

- A general library extension to NumPyro, called EinSteinVI. EinSteinVI allows SVGD to work with custom guide programs based on ELBO-within-Stein optimization. The library is compositional with NumPyro features, including support for deep learning, loss functions (ELBO, Rényi ELBO (Li & Turner, 2016), custom losses), and optimization methods, thus making it possible for EinSteinVI to grow organically with NumPyro development.

- Integration of recent developments in Stein variational inference which includes: non-linear optimization (Wang & Liu, 2019a), a wealth of kernels (Liu & Wang, 2016; 2018; Gorham & Mackey, 2017), matrix-valued kernels (Wang et al., 2019a) supporting higher-order optimization, an (experimental) update based on Stein point MCMC (Chen et al., 2019a), and factorization based on conditional independence between elements in the model (graphical kernels) (Wang et al., 2018b).

- A series of examples that demonstrate EinSteinVI and the synergy between different Stein VI techniques. The examples include a novel Stein-mixture version of the deep Markov

model (SM-DMM), Bayesian neural networks for regression. An examples that explore kernels and higher-order optimization is available in the appendix.

The paper proceeds as follows. We first present the background of SVGD in the context of our integrated implementation in Section 2. In Section 3 we introduce Stein mixtures and our algorithm ELBO-within-Stein. We discuss the general details of the implementation of EinSteinVI in Section 4. Next, we discuss related work in Section 5. In Section 6 we present various examples using EinSteinVI, and finally, we summarize our results and discuss future work in Section 7.

## 2 STEIN VARIATIONAL GRADIENT DESCENT

The core idea of SVGD is to perform inference by approximating the target posterior distribution $p(\mathbf{z}|\mathbf{x})$ by an empirical distribution $q_{\mathcal{Z}}(\mathbf{z}) = N^{-1}\sum_i \delta_{\mathbf{z}_i}(\mathbf{z})$ based on a set of particles $\mathcal{Z} = \{\mathbf{z}_i\}_{i=1}^N$. Here, $\delta_{\mathbf{x}}(\mathbf{y})$ represents the Dirac delta measure, which is equal to 1 if $\mathbf{x} = \mathbf{y}$ and 0 otherwise. One could thus see the approximating distribution $q_{\mathcal{Z}}(\mathbf{z})$ as a mixture of point estimates, each represented by a particle $\mathbf{z}_i \in \mathcal{Z}$. The idea is to minimize the Kullback-Leibler divergence $D_{\mathrm{KL}}(q_{\mathcal{Z}}(\mathbf{z}) \parallel p(\mathbf{z}|\mathbf{x}))$ between the approximation and the true posterior by iteratively updating the particles using the Stein forces:

$$\mathbf{z}_i \leftarrow \mathbf{z}_i + \epsilon S_{\mathcal{Z}}(\mathbf{z}_i)$$

where $\epsilon$ is the learning rate and $S_{\mathcal{Z}}$ denotes the Stein forces.

**The Two Forces of SVGD**   Stein VI consists of two forces which work additively under the form $S_{\mathcal{Z}} = S_{\mathcal{Z}}^+ + S_{\mathcal{Z}}^-$, where the attractive force is given by

$$S_{\mathcal{Z}}^+(\mathbf{z}_i) = \mathbb{E}_{\mathbf{z}_j \sim q_{\mathcal{Z}}(\mathbf{z})}[k(\mathbf{z}_i, \mathbf{z}_j)\nabla_{\mathbf{z}_j} \log p(\mathbf{z}_j|\mathbf{x})]$$

and the repulsive force by

$$S_{\mathcal{Z}}^-(\mathbf{z}_i) = \mathbb{E}_{\mathbf{z}_j \sim q_{\mathcal{Z}}(\mathbf{z})}[\nabla_{\mathbf{z}_j} k(\mathbf{z}_i, \mathbf{z}_j)].$$

Here $k : \mathbb{R}^d \times \mathbb{R}^d \to \mathbb{R}$ is a kernel. The attractive force can be seen as pushing the particles towards the modes of the true posterior distribution, smoothed by some kernel. For an example of a kernel, consider the radial basis function (RBF) kernel $k(\mathbf{z}_i, \mathbf{z}_j) = \exp\left(-\frac{1}{h} \parallel \mathbf{z}_i - \mathbf{z}_j \parallel_2^2\right)$ with bandwidth parameter $h$, chosen as $\frac{1}{\log n}\mathrm{med}(\mathbf{z})$.

The repulsive force moves particles away from each other, ensuring that they do not collapse to the same mode. For example, with the RBF kernel, the repulsive force becomes $\mathbb{E}_{\mathbf{z}_j \sim q_{\mathcal{Z}}(\mathbf{z})}[-\exp\left(-\frac{1}{h} \parallel \mathbf{z}_i - \mathbf{z}_j \parallel_2^2\right) \frac{2}{h}\sum_\ell (\mathbf{z}_{i\ell} - \mathbf{z}_{j\ell})]$, which has high kernel values for particles that are close, causing them to repel.

SVGD works with unnormalized distributions $p$ as the normalization constant becomes additive in the log-posterior $\log p(\mathbf{z}_i|\mathbf{x}) = -\log Z + \log p(\mathbf{x}|\mathbf{z}) + \log p(\mathbf{z})$ and is thus not required for the calculation of the gradient. This property is desirable as normalizing $p$ is often computationally expensive.

**Non-linear Stein**   In non-linear Stein (Wang & Liu, 2019a), the repulsive force can be scaled by a factor $\lambda$, resulting in $S_{\mathcal{Z}} = S_{\mathcal{Z}}^+ + \lambda S_{\mathcal{Z}}^-$. This approach is useful when dealing with multimodal distributions. It is also useful in cases where the repulsive force vanishes compared to the likelihood, which happens for large datasets ($\mathcal{X}$). Scaling the repulsive force by a constant $\lambda = c(|\mathcal{X}|)$ proportional (e.g. $c = 0.1$ or $c = 0.01$) to the size of the dataset $|\mathcal{X}|$ addresses this issue and can be chosen by cross-validation on a subset of the data.

**Matrix-valued kernels**   The choice of kernels can be extended to matrix-valued ones (Wang et al., 2019a), $K : \mathbb{R}^d \times \mathbb{R}^d \to \mathbb{R}^{d \times d}$, in which case the Stein forces become

$$S_{\mathcal{Z}}^+(\mathbf{z}_i) = \mathbb{E}_{\mathbf{z}_j \sim q_{\mathcal{Z}}(\mathbf{z})}[K(\mathbf{z}_i, \mathbf{z}_j)\nabla_{\mathbf{z}_j} \log p(\mathbf{z}_j|\mathbf{x})]$$

and

$$S_{\mathcal{Z}}^-(\mathbf{z}_i) = \mathbb{E}_{\mathbf{z}_j \sim q_{\mathcal{Z}}(\mathbf{z})}[K(\mathbf{z}_i, \mathbf{z}_j)\nabla_{\mathbf{z}_j}]$$

where the standalone del $\nabla_{\mathbf{z}_j}$ in the repulsive force represents the vector $\left( \frac{\partial}{\partial z_{j,1}}, \ldots, \frac{\partial}{\partial z_{j,d}} \right)$. This results in

$$(K(\mathbf{z}_i, \mathbf{z}_j)\nabla_{\mathbf{z}_j})_\ell = \sum_k \nabla_k K_{\ell,k}(\mathbf{z}_i, \mathbf{z}_j).$$

The advantage of matrix-valued kernels is that they allow preconditioning[2] using the Hessian or Fisher Information matrix, which can capture local curvature and thus achieve better optima and convergence rate than standard SVGD. Furthermore, it is easy to represent graphical kernels (Wang et al., 2018a) using matrix kernels, e.g. $K = \mathrm{diag}(\{K^{(\ell)}\}_\ell)$ where the set of variables are partitioned with each their own local kernel $K^{(\ell)}$.

## 3  STEIN MIXTURES

The Stein-mixture was proposed by Nalisnick & Smyth (2017) to resolve the representation issue for SVGD when the target distribution has high dimensionality. For SVGD, the number of particles needed to represent a distribution adequately grows exponentially with its dimensionality. As the computational complexity for the update rule in SVGD is quadratic in the number of particles, the exponential growth quickly becomes computationally intractable.

Stein-mixtures are hierarchical variational models (Ranganath et al., 2016b) such that the posterior of the second tier variational distribution is $q(\mathbf{z}|X) = \frac{1}{N} \sum_{k=1}^N \delta(\mathbf{z}_k)$. The guide is the joint model $q(\boldsymbol{\theta}, \mathbf{z}) = q(\boldsymbol{\theta}|\mathbf{z})q(\mathbf{z})$, with $q(\boldsymbol{\theta}|\mathbf{z})$ the first tier distribution and $q(\mathbf{z})$ its prior. The second tier posterior $q(\mathbf{z}|X)$ is optimized using SVGD, so that the marginal posterior is $q(\boldsymbol{\theta}; \mathbf{z}) = \frac{1}{N} \sum_{k=1}^N q(\boldsymbol{\theta}|\mathbf{z})$, a restricted mixture.

Nalisnick & Smyth (2017) showed that the Stein force for a mixture approximation of the posterior $p(\mathbf{z}|Z)$ is given by

$$\begin{aligned} S_{\mathcal{Z}}(\mathbf{z}_i) = {} & \mathbb{E}_{\mathbf{z}_j \sim q_{\mathcal{Z}}(\mathbf{z})} \left[ k(\mathbf{z}_i, \mathbf{z}_j) \mathbb{E}_{\boldsymbol{\theta}_l \sim q(\theta|\mathbf{z}_j)} \left[ p(X, \boldsymbol{\theta}_l)/q(\boldsymbol{\theta}_l|\mathbf{z}_j) \right] \right] + \\ & + \mathbb{E}_{\mathbf{z}_j \sim q_{\mathcal{Z}}(\mathbf{z})} \left[ k(\mathbf{z}_i, \mathbf{z}_j) \nabla_{\mathbf{z}_j} \log q(\mathbf{z}_j) \right] + \\ & + S_{\mathcal{Z}}^-(\mathbf{z}_i), \end{aligned}$$

where $\log \nabla_{\mathbf{z}_j} q(\mathbf{z}_j)$ acts as a regularizer on the variational parameters which they assume is sufficiently small to be dropped. To evaluate $\mathbb{E}_{\mathbf{z}_j \sim q_{\mathcal{Z}}(\mathbf{z})} \left[ p(X, \theta)/q(\theta|\mathbf{z}_j) \right]$, Nalisnick & Smyth (2017) use DNCP and importance weighted Monte Carlo gradients to obtain the black-box update,

$$S_{\mathcal{Z}}(\mathbf{z}_i) = \mathbb{E}_{\mathbf{z}_j \sim q_{\mathcal{Z}}(\mathbf{z})} \left[ k(\mathbf{z}_i, \mathbf{z}_j) \sum_{s=1}^S \tilde{w}_s \nabla_{\mathbf{z}_j} \log \left( \frac{p(X, \hat{\boldsymbol{\theta}}_s)}{q(\hat{\boldsymbol{\theta}}_s|\mathbf{z}_j)} \right) \right] + S_{\mathcal{Z}}^-(\mathbf{z}_i), \tag{1}$$

where $\tilde{w}_s$ is an importance weight for sample $\hat{\boldsymbol{\theta}}_s = q(\mathbf{z}_j, \boldsymbol{\xi})$, $\boldsymbol{\xi} \sim p_0$ (i.e. the guide).

**ELBO-within-Stein**  In ELBO-within-Stein, we replace the weighted average over $S$ samples in Equation (1) by a single loss ($\mathcal{L}$). If we choose $\mathcal{L}$ to be an $f$-divergence, such as the ELBO, we can reduce the loss bias by averaging multiple samples. However, ELBO-within-Stein only computes one gradient of $\mathcal{L}$ per particle regardless of the number of samples we use to estimate $\mathcal{L}$. This is different from Nalisnick & Smyth (2017) who estimate $\mathbb{E}_{\boldsymbol{\theta}_l \sim q(\theta|\mathbf{z}_j)} \left[ p(X, \boldsymbol{\theta}_l)/q(\boldsymbol{\theta}_l|\mathbf{z}_j) \right]$ by averaging over gradients, therefore computes a gradient and an importance weighting for each DNCP sample rather than per particle. The difference makes ELBO-within-Stein computationally cheaper as the complexity of adding a gradient is $\mathcal{O}(n)$ whereas adding a sample to estimate $\mathcal{L}$ is $\mathcal{O}(1)$. The Stein force in ELBO-within-Stein is given by

$$S_{\mathcal{Z}}(\mathbf{z}_i) = \mathbb{E}_{\mathbf{z}_j \sim q_{\mathcal{Z}}(\mathbf{z})} \left[ k(\mathbf{z}_i, \mathbf{z}_j) \nabla_{\mathbf{z}_j} \mathcal{L}(z_j) \right] + S_{\mathcal{Z}}^-(\mathbf{z}_i) \tag{2}$$

where $\mathcal{L}$ is the ELBO.

---

[2]Using preconditioner matrix $Q$, such that $Q^{-1}M$ has a lower condition number than $M$.

## 4 COMPOSITIONAL IMPLEMENTATION USING NUMPYRO

EinSteinVI integrates with the existing NumPyro API by adding the Stein VI interface, which closely mimicks NumPyro's SVI interface. Mimicking the SVI interface makes programs that use SVI in NumPyro trivial to convert to EinSteinVI (see Figure 1).

Below, we discuss the key features of EinSteinVI, which include re-initializable guides, EinSteinVI's core algorithm, and the new kernel interface.

### 4.1 RE-INITIALIZABLE GUIDES

The Stein VI interface requires that the initialization is different for each parameter in an inference program. The reason is that different Stein particles need to be initialized to different values in order for optimization to work correctly and to avoid that all particles collapse into the posterior mode.

To support re-initializable guides, we provide the `ReinitGuide` interface, which requires implementing a function `find_params` that accepts a list of random number generator (RNG) keys in addition to the arguments for the guide and returns a set of freshly initialized parameters for each RNG key.

The `WrappedGuide` class provides a guide written as a function. `WrappedGuide` makes a callable guide re-initializable. It works by running the provided guide multiple times and reinitializing the parameters using NumPyro's interface as follows:

- `WrappedGuide` runs the guide transforming each parameter to unconstrained space.
- It replaces the values of the parameters with values provided by a NumPyro initialization strategy, e.g., `init_to_uniform`($r$), which initializes each parameter with a uniform random value in the range $[-r; r]$.
- It saves the parameter values for each particle and the required inverse transformations to constrained space to run the model correctly.

We also allow parameters without reinitialization in order to support neural network libraries like `stax` [3] that have their own initializers.

The Stein VI interface will correctly wrap the guide during initialization, so from a user perspective the syntax for guides follows the API of SVI.

### 4.2 STEIN VI IN NUMPYRO

The integration of Stein VI into NumPyro requires handling transformations between the parameter representation of NumPyro[4] and the vectorized Stein particles that EinSteinVI operates on. For this, we rely on `Jax PyTrees`[5] which converts back and forth between Python collections and a flattened vectorized representation.

Algorithm 1 shows the core algorithm of EinSteinVI. EinSteinVI updates the standard variational model parameters $\phi$ and guide parameters $\psi$ by averaging the loss over the Stein particles. For the Stein parameters, the process is more elaborate. First, we convert the set of individual parameters to a monolithic vector-encoded particle using `Jax PyTrees`. The monolithic particle represents the particles as a flattened and stacked `Jax` array. Then we compute a kernel based on the vector-encoded Stein particle; this is delegated to the kernel interface as the computation is kernel-dependent.

We apply `Jax`'s `vmap` operator (Frostig et al., 2018; Phan et al., 2019) to compute the Stein forces for each particle in a vectorized manner. This is done in unconstrained space so the Stein force must the corrected by the Jacobian of the bijection between constrained and unconstrained space. Doing this directly on the `Jax` on the monolithic particle incurs a massive memory overhead in the adjoint. However, as NumPyro registers a bijection for each distribution parameter we can eliminate the overhead by computing the Jacobian on the `Jax` representations of the individual parameters. The operation is embarrassingly parallel and so we again use a `vmap` operator with a nested `tree_map` to

---

[3] `https://jax.readthedocs.io/en/latest/jax.experimental.stax.html`
[4] A dictionary mapping parameters to their values, which can be arbitrary Python type
[5] `https://jax.readthedocs.io/en/latest/pytrees.html`

compute the desired Jacobians. Note that Algorithm 1 presents how EinSteinVI works with scalar kernels and does not account for the different features presented in Section 2.

Finally, we convert the monolithic Stein particle to their non-vectorized dictionary-based form and return the expected changes for standard- and Stein-parameters.

**Input:** Classical VI parameters $\phi$ and $\psi$, Stein parameters $\{\boldsymbol{\theta}_i\}_i$, model $p_\phi(\mathbf{z}, \mathbf{x})$, guide $q_{\boldsymbol{\theta},\psi}(\mathbf{z})$, loss $\mathcal{L}$, kernel interface KI.
**Output:** Parameter changes based on classical VI ($\Delta\phi$, $\Delta\psi$) and Stein VI forces ($\{\Delta\boldsymbol{\theta}_i\}_i$).
$\quad$ **procedure** EINSTEIN($\phi$, $\psi$, $\{\boldsymbol{\theta}_i\}_i$, $p_\phi$, $q_{\boldsymbol{\theta},\psi}$)
$\quad\quad\quad \Delta\phi \leftarrow \mathbb{E}_{\boldsymbol{\theta}}[\nabla_\phi \mathcal{L}(p_\phi, q_{\boldsymbol{\theta},\psi})]$
$\quad\quad\quad \Delta\psi \leftarrow \mathbb{E}_{\boldsymbol{\theta}}[\nabla_\psi \mathcal{L}(p_\phi, q_{\boldsymbol{\theta},\psi})]$
$\quad\quad\quad \{\mathbf{a}_i\}_i \leftarrow \text{PyTreeFlatten}(\{\boldsymbol{\theta}_i\}_i)$
$\quad\quad\quad k \leftarrow \text{KI}(\{\mathbf{a}_i\}_i)$

$\quad\quad\quad$ **procedure** EINSTEINFORCES($\mathbf{a}_i$) $\qquad\qquad$ ▷ Calculate forces per particle for higher-order vmap function.
$\quad\quad\quad\quad\quad \boldsymbol{\theta}_i \leftarrow \text{PYTREERESTORE}(\mathbf{a}_i)$
$\quad\quad\quad\quad\quad \Delta\mathbf{a}_i \leftarrow \sum_{\mathbf{a}_j} k(\mathbf{a}_j, \mathbf{a}_i)\nabla_{\mathbf{a}_i}\mathcal{L}(p_\phi, q_{\boldsymbol{\theta}_i,\psi}) + \nabla_{\mathbf{a}_i}k(\mathbf{a}_j, \mathbf{a}_i)$
$\quad\quad\quad\quad\quad$ **return** $\Delta\mathbf{a}_i$
$\quad\quad\quad$ **end procedure**

$\quad\quad\quad \{\Delta\mathbf{a}_i\}_i \leftarrow \text{VMap}(\{\mathbf{a}_i\}_i, \text{EINSTEINFORCES})$
$\quad\quad\quad \{\Delta\boldsymbol{\theta}_i\}_i \leftarrow \text{PYTREERESTORE}(\{\Delta\mathbf{a}_i\}_i)$
$\quad\quad\quad$ **return** $\Delta\phi, \Delta\psi, \{\Delta\boldsymbol{\theta}_i\}_i$
$\quad$ **end procedure**

**Algorithm 1:** EinSteinVI

### 4.3 KERNEL INTERFACE

The Kernel interface is straightforward. To extend the interface users must implement the `compute` function, which accepts as input the current set of particles, the mapping between model parameters and particles, and the loss function $\mathcal{L}$ and returns a differentiable kernel $k$. All kernels are currently static, but the interface could be extended to stateful kernels, allowing conjugate gradients or quasi-Newton optimization. Appendix A gives the complete list of kernels in EinSteinVI. We are planning to extend EinSteinVI with probability product kernels Nalisnick & Smyth (2017), which take into account the information geometry of the first tier variational distributions in Stein mixtures.

## 5 RELATED WORK

There has been a proliferation of deep probabilistic programming languages (Ge et al., 2018; Bingham et al., 2019; Salvatier et al., 2016; Tran et al., 2016; Cusumano-Towner et al., 2019; Dillon et al., 2017) based on tensor frameworks featuring automatic differentiation and supporting various inference techniques. However, only `PyMC3` (Salvatier et al., 2016) includes inference using SteinVI.

In `PyMC3` the inference techniques that use Stein's method include SVGD with a scalar RBF-kernel, amortized SVGD (Wang et al., 2016; Feng et al., 2017), and Operator Variational Inference (OVI) (Ranganath et al., 2016a).

SVGD in `PyMC3` manipulates particles to directly capture the target distribution (Liu & Wang, 2016). In EinSteinVI, SVGD is a special case of Stein mixtures where the guides are point mass distributions. Because we have guide programs, EinSteinVI allows arbitrary computations to transform random variables in the guide. This is not possible with SVGD in `PyMC3`.

Amortized SVGD (Feng et al., 2017) trains a stochastic network to draw samples from a target distribution. The network is iteratively adjusted so that the output changes in the direction of the Stein variational gradient (the same gradient as used in SVGD). In comparison, EinSteinVI transports a fixed set of particles (each parameterizing a guide) to the target distribution. Amortized SVGD is not in EinSteinVI as its extension to arbitrary guides is an open problem.

```python
def model():
    sample('x', NormalMixture(jnp.array([1 / 3, 2 / 3]),
                              jnp.array([-2.0, 2.0]),
                              jnp.array([1.0, 1.0])))
```

(a) 1D Gaussian mixture model

```python
svi = SVI(
        model,
        AutoNormal(model),
        Adagrad(step_size=1.0),
        Trace_ELBO()
    )

results = svi.run(rng_key,
                  num_iterations)
```

```python
stein = SteinVI(
            model,
            AutDelta(model),
            Adagrad(step_size=1.0),
            Trace_ELBO(),
            RBFKernel(),
        )

results = stein.run(rng_key,
                    num_iterations)
```

(b) SVI

(c) SVGD with EinSteinVI

Figure 1: 1D Gaussian mixture model in NumPyro.

OVI optimizes operator objectives, which take functions of functions to a non-negative number. Ranganath et al. (2016a) include an operator objective based on the Langevin-Stein operator (Anastasiou et al., 2021). This is the same operator used for the kernelized Stein discrepancy (Liu et al., 2016) which also underlies SVGD. Unlike EinSteinVI, Amortized SVDG and SVGD, OPV is not a particle-based method.

## 6 EXAMPLES

We illustrate the features of EinSteinVI on two toy examples, namely a 1D mixture of Gaussians below, and 2D mixtures of Gaussian in Appendix B. We demonstrate EinSteinVI on real-world examples and show that EinSteinVI tends to outperform alternative methods. These examples include regression with Bayesian neural networks and deep Markov models. All experimental code, notebook tutorials and the EinSteinVI package itself are available through an Anonymized URL.

**1D Gaussian mixture** To demonstrate the two modes of VI with EinSteinVI, we consider the 1D Gaussian mixture $1/3\mathcal{N}(-2, 1) + 2/3\mathcal{N}(2, 1)$ (see Figure 1 and Figure 2). The Gaussian target mixture is bi-modal and well-suited for the nonparametric nature of SVGD and Stein mixtures. Figure 2 shows that both SVGD and the Stein-mixture naturally capture the bi-modality of the target distribution, compared to SVI with a simple Gaussian guide. Note the reduction in particles required to estimate the target when using Stein mixtures compared to SVGD. Also, note that the Stein-mixture overestimates the variance and slightly pertubates the locations. The error seen at the right mode for the Stein-mixture with two particles is due to the uniform weighting of the particles in SVGD (the target posterior is approximated with an empirical distribution, see Section 2), and as such is algorithmic. The Stein-mixture will therefore not be able to exactly capture the mixing components of a target mixture model with one particle per component. However, with more particles the mixture can be approximated better as demonstrated using three particles.

**Bayesian Neural Networks** We compare SVGD and non-linear Stein in EinSteinVI with the implementation of SVGD by Liu & Wang (2016)[6] (without model selection), and amortized SVGD (using the Theano backend[7], Al-Rfou et al. (2016)) on Bayesian neural networks (BNN) for regression. The PyMC3 documentation clearly states amortized SVGD is experimental and is not suggested to be used. We include its experimental results in Appendix B for completeness and note that our findings reflect the warning in the PyMC3 documentation. We use an RBF-kernel in EinSteinVI for fair comparison with Liu & Wang (2016) and PyMC3. For non-linear Stein we determined the best repulsion factor $\lambda^*$ by a grid search for $\lambda \in \{10^{-2}, 10^{-1}, 10^1, 10^2\}$. For SVGD PyMC3 we use a

---

[6]https://github.com/dilinwang820/Stein-Variational-Gradient-Descent/blob/master/python/bayesian_nn.pyy

[7]PyMC4, which uses Jax as its backend, does not include SVGD at the time of writing.

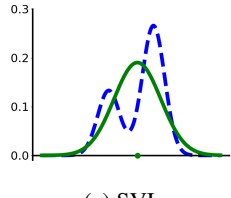 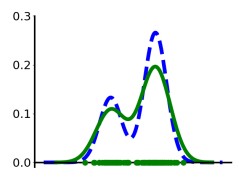 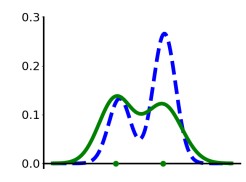 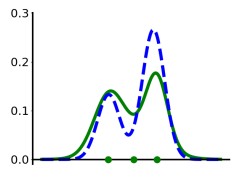

(a) SVI     (b) SVGD (RBF kernel)     (c) Two particle Stein-mixture (linear kernel)     (d) Three particle Stein-mixture (RBF kernel)

Figure 2: The blue dashed line is the target pdf, while the solid green line is the density of the particles. We estimate the particle density for SVGD with Gaussian kernel density estimation. We use 100 particles for SVGD, and two or three particles for the Stein-mixture. SVI uses a Gaussian guide.

Table 1: Average test RMSE and time for inference for the UCI regression benchmarks.

| | Test RMSE | | | | Time | |
|---|---|---|---|---|---|---|
| | **EinStein** | | **PyMC3** | **Liu & Wang (2016)** | **EinStein** | **PyMC3** |
| Dataset | SVGD | NL-Stein | SVGD | SVGD | SVGD | SVGD |
| Boston | $2.610 \pm 0.044$ | $2.492 \pm 0.028$ | $6.059 \pm 0.244$ | $2.990 \pm 0.013$ | $5.758s \pm 0.048s$ | $1m6.396s \pm 0.301s$ |
| Concrete | $4.175 \pm 0.025$ | $4.782 \pm 0.047$ | $6.255 \pm 0.390$ | $5.927 \pm 0.021$ | $5.602s \pm 0.048s$ | $1m5.845s \pm 0.278s$ |
| Energy | $0.321 \pm 0.004$ | $0.375 \pm 0.007$ | $3.494 \pm 1.500$ | $1.251 \pm 0.017$ | $5.562s \pm 0.053s$ | $1m6.175s \pm 0.41s$ |
| Kin8nm | $0.073 \pm 0.000$ | $0.076 \pm 0.000$ | $0.164 \pm 0.040$ | $0.115 \pm 0.001$ | $6.141s \pm 0.04s$ | $2m40.106 \pm 0.298s$ |
| Naval | $0.001 \pm 0.000$ | $0.001 \pm 0.000$ | $0.029 \pm 0.015$ | $0.008 \pm 0.001$ | $6.51s \pm 0.048s$ | $3m51.05s \pm 0.385s$ |
| Power | $3.952 \pm 0.004$ | $3.907 \pm 0.006$ | $5.389 \pm 2.219$ | $4.215 \pm 0.002$ | $6.314s \pm 0.054s$ | $2m51.654s \pm 0.566s$ |
| Protein | $4.509 \pm 0.006$ | $4.654 \pm 0.006$ | $5.921 \pm 2.559$ | $4.846 \pm 0.002$ | $9.436s \pm 0.032s$ | $11m11.9s \pm 0.399s$ |
| Wine | $0.599 \pm 0.006$ | $0.596 \pm 0.005$ | $0.769 \pm 0.212$ | $0.604 \pm 0.001$ | $5.614s \pm 0.052s$ | $1m7.767s \pm 0.286s$ |
| Yacht | $0.531 \pm 0.009$ | $0.246 \pm 0.013$ | $8.341 \pm 3.744$ | $0.94 \pm 0.03$ | $5.519s \pm 0.07s$ | $1m4.748s \pm 0.392s$ |
| Year | $8.662 \pm 0.002$ | $8.701 \pm 0.002$ | $135.316$ | $8.895 \pm 0.001$ | $1m37.569s \pm 0.07s$ | $4h19m36s$ |

temperature[8] of ten. Like Liu & Wang (2016) we use a BNN with one hidden layer of size fifty and `RELU` activation. We put a $Gamma(1, 0.1)$ prior on the precision of the neurons and the likelihood. For all versions, we use 100 particles and 2000 iterations. We use a mini-batch of 1000 for Year and 100 for the rest. All measurements are repeated ten times and obtained on a GPU[9] for EinSteinVI and `PyMC3`, except for Year with `PyMC3` which was only run once. We do not report times for Liu & Wang (2016) because only the CPU version of their code could be executed without irresolvable issues.

Table 1 shows the performance in terms of the average root mean squared error (RMSE) on the test set. We find that EinSteinVI achieves significantly better RMSE than Liu & Wang (2016) and that both systems outperform SVGD in `PyMC3`. The times in Table 1 measure how long it takes to infer parameters. Table 1 excludes the first run for two reasons: i) that run will fill the GPU caches, and ii) `Jax` will trace the programs. As a result, this run is more costly than subsequent ones. The times for the first run are given in Table 2. By running EinSteinVI for more iterations, we can amortize the initial cost.

**Stein Mixture Deep Markov Model**    Music generation requires a model to learn complex temporal dependencies to achieve local consistency between notes. The Stein-mixture deep Markov model (SM-DMM) is a deep Markov model that uses a mixture of Stein particles to estimate distributions over model parameters. We consider a vectorized version of the DMM (Jankowiak & Karaletsos, 2019) for the generation of polyphonic music using the JSB chorales data set.

The SM-DMM model consists of two feed-forward neural (FNN) networks. The *Transition* network takes care of the conditional dependencies between subsequent latent states in the Markov chain. It consist of two layers with hidden dimension 200 and ReLU activation on the first layer and sigmoid

---

[8]We found the performance to be very sensitive to the choice of temperature.

[9]Quadro RTX 6000 with Cuda V11.4.120

Table 2: Time for first repetition with EinStein for UCI regression benchmarks.

| Dataset | Boston | Concrete | Energy | Kin8nm | Naval | Power | Protein | Wine | Yacht | Year |
|---|---|---|---|---|---|---|---|---|---|---|
| Time | 41.665s | 41.642s | 41.591s | 43.592s | 44.2570s | 44.058s | 47.87s | 41.9s | 41.409s | 2m18.19s |

Table 3: Test negative log- likelihood (lower is better) on Polyphonic Music Generation (JSB) dataset. Baseline results from Krishnan et al. (2016). ISN-DMM and ISN-DMM-Aug (Krishnan et al., 2016), TSBN and HMSBN (Gan et al., 2015)

|        | ISN-DMM | ISN-DMM-Aug | HMSBN  | TSBN | SM-DMM  |
|--------|---------|-------------|--------|------|---------|
| NLL (a) | 6.926   | 6.773       | 8.0473 | -    | -45.983 |
| NLL (b) | 6.856   | 6.692       | 7.9970 | 7.48 | -46.066 |

on the second. The *Emitter* network is a three layers FNN which produces a likelihood at each time step using the current latent state. The layers have hidden dimensions 100, 100, and 88, respectively. The variational distribution is of the form $\prod_{n=1}^{N} q(z_{1:T_n}^n | f(X_{1:T_n}))$ where the parametrized feature function $f_{1:T_n}$ is a one layered gated recurrent unit (Chung et al., 2014), with hidden dimension 150. The total number of parameters in the DMM is 128,654, so that with five particles EinSteinVI is optimizing 643,270 parameters for the SM-DMM model.

We train the SM-DMM using the Adam optimizer (Kingma & Ba, 2014) with a learning rate of $10^{-5}$, using an RBF kernel and five Stein particles for four thousand epochs on the polyphonic music generation dataset JSB chorals (Boulanger-Lewandowski et al., 2012). We follow Krishnan et al. (2016) and report two version NLL of a) $\frac{\sum_{i=1}^{N} -p(\mathbf{x}_i|\boldsymbol{\theta})}{\sum_{i=1}^{N} T_i}$ and b) $\frac{1}{N} \sum_{i=1}^{N} \frac{-p(\mathbf{x}_i|\boldsymbol{\theta})}{T_i}$, where $T_i$ is the length of the $i$th sequence. In Table 3 we report NLL(a) and NLL(b) on a held-out test set of JSB. Compared to baseline methods SM-DMM achieves a significant improvement using Stein mixtures. We see similar improvements in the test ELBO, Jankowiak & Karaletsos (2019) reports a test ELBO of -6.82 nats on the JSB dataset for their approach, SM-DMM with EinSteinVI achieves an test ELBO of *45.10* (higher is better).

## 7 SUMMARY

EinSteinVI provides the latest techniques for Stein VI as an extension to Numpyro. Our results indicate that the library is substantially faster and more expressive than other available libraries for Stein VI. EinSteinVI provides a familiar and efficient interface for practitioners working with the Pyro/NumPyro PPL and provides a unified code base to researchers for benchmarking new developments in Stein VI.

Possible further extensions include (a) supporting updates inspired by Stein Points (Chen et al., 2018; 2019b), (b) extending the kernel interface with probability product kernels (Jebara et al., 2004) to account for information geometry when using Stein-mixtures, (c) adding full support for NumPyro's automatic enumeration features (Obermeyer et al., 2019) when guides are used, and (d) stateful matrix kernels for conjugate gradient or quasi-Newton optimization. Stein Points updates could handle cases where initialization of particles is sub-optimal by initializing from a Stein point MCMC chain and interleave MCMC updates to get better mode hopping properties.

ACKNOWLEDGEMENTS

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

## A  APPENDIX

| Kernel | Definition | Detail | Type | Reference |
|---|---|---|---|---|
| Radial Basis Function (RBF) | $\exp(\frac{1}{h} \parallel \mathbf{x} - \mathbf{y} \parallel_2^2)$ | | scalar | Liu & Wang (2016) |
| | $\exp(\frac{1}{h}(\mathbf{x} - \mathbf{y}))$ | | vector | Pyro[10] |
| Inverse Multi-Quadratic (IMQ) | $(c^2 + \parallel \mathbf{x} - \mathbf{y} \parallel_2^2)^\beta$ | $\beta \in (-1, 0)$ and $c > 0$ | scalar | Gorham & Mackey (2017) |
| Random Feature Expansion | $\mathbb{E}_{\mathbf{w}}[\phi(\mathbf{x}, \mathbf{w})\phi(\mathbf{y}, \mathbf{w})]$ | $\phi(\mathbf{x}, \mathbf{w}) = \sqrt{2}\cos(\frac{1}{h}\mathbf{w}_1^\top \mathbf{x} + \mathbf{w}_0)$ where $\mathbf{w}_0 \sim \mathrm{Unif}(0, 2\pi)$ and $\mathbf{w}_1 \sim \mathcal{N}(0, 1)$ | scalar | Liu & Wang (2018) |
| Linear | $\mathbf{x}^\top \mathbf{y} + 1$ | | scalar | Liu & Wang (2018) |
| Mixture | $\sum_i w_i k_i(\mathbf{x}, \mathbf{y})$ | $\{k_i\}_i$ individual kernels, weights $w_i$ | scalar, vector, matrix | Liu & Wang (2018) |
| Scalar-based Matrix | $k(\mathbf{x}, \mathbf{y})\boldsymbol{I}$ | $k$ scalar-valued kernel | matrix | Wang et al. (2019a) |
| Vector-based Matrix | $\mathrm{diag}(k(\mathbf{x}, \mathbf{y}))$ | $k$ vector-valued kernel | matrix | Wang et al. (2019a) |
| Graphical | $\mathrm{diag}(\{K^{(\ell)}(\mathbf{x}, \mathbf{y})\}_\ell)$ | $\{K^{(\ell)}\}_\ell$ matrix-valued kernels, each for a unique partition of latent variables | matrix | Wang et al. (2019a) |
| Constant Pre-conditioned | $\boldsymbol{Q}^{-\frac{1}{2}} K(\boldsymbol{Q}^{\frac{1}{2}}\mathbf{x}, \boldsymbol{Q}^{\frac{1}{2}}\mathbf{y})\boldsymbol{Q}^{-\frac{1}{2}}$ | $K$ is an inner matrix-valued kernel and $\boldsymbol{Q}$ is a preconditioning matrix like the Hessian $-\nabla_{\bar{\mathbf{z}}}^2 \log p(\bar{\mathbf{z}}|\mathbf{x})$ or Fischer information $-\mathbb{E}_{\mathbf{z} \sim q_{\mathcal{Z}}(\mathbf{z})}[\nabla_{\mathbf{z}}^2 \log p(\mathbf{z}|\mathbf{x})]$ matrices | matrix | Wang et al. (2019a) |
| Anchor Point Preconditioned | $\sum_{\ell=1}^m K_{\boldsymbol{Q}_\ell}(\mathbf{x}, \mathbf{y})w_\ell(\mathbf{x})w_\ell(\mathbf{y})$ | $\{\mathbf{a}_\ell\}_{\ell=1}^m$ is a set of anchor points, $\boldsymbol{Q}_\ell = \boldsymbol{Q}(\mathbf{a}_\ell)$ is a preconditioning matrix for each anchor point, $K_{\boldsymbol{Q}_\ell}$ is an inner kernel conditioned using $\boldsymbol{Q}_\ell$, and $w_\ell(\mathbf{x}) = \mathrm{softmax}_\ell(\{\mathcal{N}(\mathbf{x}|\mathbf{a}_{\ell'}, \boldsymbol{Q}_{\ell'}^{-1})\}_{\ell'})$ | matrix | Wang et al. (2019a) |

## B  EXPERIMENTS

**Star Distribution**  To illustrate the kernels included in EinSteinVI, we approximate the "star distribution", follow Wang et al. (2019b) (see Figure 3). The star distribution is constructed as a 2D Gaussian mixture,

$$p(x) = \frac{1}{K}\sum_{k=1}^K \mathcal{N}(x|\mu_k, \Sigma_k) \quad \mu_1 = [0, 1.5], \mu_k = U_k\mu_1 \quad \Sigma_1 = \mathrm{diag}([1, \frac{1}{100}]), \Sigma_k = U_k\Sigma_1 U_k$$

where $U_k$ is a rotation matrix given by

$$U_k = \begin{bmatrix} \cos(\theta) & -\sin(\theta) \\ \sin(\theta) & \cos(\theta) \end{bmatrix}, \quad (k \in [1, .., K])\theta = 2k\pi/K.$$

We use fifty particles and Adagrad with the learning rate that yields the best Maximum Mean Discrepancy (MMD) (Gretton et al., 2012) with a RBF kernel. To compute the MMD, we use 10,000

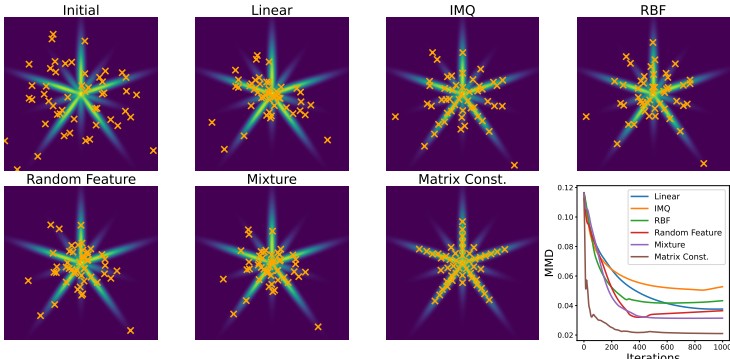

Figure 3: Particle positions for EinSteinVI with different kernels after 1000 iterations with a point mass guide, starting from particle positions given in the upper left frame labelled Initial. The MMD, bottom right frame, is evaluated using the RBF kernel.

Table 4: Average test RMSE and time for inference for the UCI regression benchmarks with amortized SVGD.

|  | RMSE | Time |
|---|---|---|
| Boston | $87230.469 \pm 54219.534$ | $5m46s \pm 3s$ |
| Concrete | $2458.250 \pm 754.653$ | $6m3s \pm 2s$ |
| Energy | $2458.250 \pm 754.653$ | $5m45s \pm 3s$ |
| Kin8nm | $302.218 \pm 43.818$ | $19m54s \pm 5s$ |
| Naval | $0.386 \pm 0.128$ | $28m39s \pm 3s$ |
| Power | $3333.796 \pm 517.241$ | $21m38s \pm 6s$ |
| Protein | $64.791 \pm 12.181$ | $1h32m28s \pm 1m1s$ |
| Wine | $12.929 \pm 6.202$ | $7m21s \pm 3s$ |
| Yacht | $2822.983 \pm 944.449$ | $5m35s \pm 4s$ |

samples from the 2D gaussian mixture. Keeping these choices fixed, we vary the type of kernel, using a point mass distribution as guide.

We consider the scalar ($\mathbb{R}^d \to \mathbb{R}$) linear kernel, the inverse multiquadric (IMQ) kernel, the RBF kernel and the random feature kernel; details are given in appendix A. In addition, we use a mixture kernel, which is a uniform mixture of the linear and the random feature kernels. We also use the matrix version of the RBF kernel with a constant preconditioning, using the Hessian matrix for preconditioning. We see that including curvature information by means of the Hessian leads to faster convergence in EinSteinVI and that the matrix RBF with preconditioning yields the lowest MMD.

**Bayesian Neural Network**  The amortized SVGD in `PyMC3` is an experimental implementation and the documentation clearly discourages its use. For completeness we include the performance here. The temperature and learning rate were determined by a grid search on the Boston data set. All measurements are repeated five times. In Table 4 we report test RMSE and time the the UCI regression benchmark. We did not include the Year data set as it takes over 21 hours to run. The RMSE for amortized SVGD is poor and highly variable across all data sets, except Naval. However, even for Naval the RMSE is two orders of magnitude greater than SVGD in EinSteinVI and Liu & Wang (2016).

