# OpenReview forum: "EinSteinVI: General and Integrated Stein Variational Inference"
_ICLR.cc/2022/Conference — ICLR 2022 Submitted_

### Official Review · Reviewer_KQPj · 2021-10-26

**Correctness:** 4
**Technical Novelty And Significance:** 1
**Empirical Novelty And Significance:** 1
**Recommendation:** 5
**Confidence:** 3

**Main Review:**

It is not clear what is the difference between EinSteinVI and the Stein VI methods included in Edward and PyMC3.

Th experiments are extensive, considering different prob. models and problems. The computational time of each method is recorded comparing favorably for the implemented one.

The paper compares favorably with competing software for prob. programing languages such as PyMC3. However, it does not compare results with Edward. This questions the significance of the results.

The novelty of the paper is low since it does not describe any new method. It only consists in an implementation of an already known method to be used within NumPyro for prob. programming.

My main concern with the paper hence the lack of novelty, although some practitioners may consider the implementation interesting and useful.



**Summary Of The Paper:**


This paper describes an implementation of EinsteinVI, an improvement over Stein VI, a method that can be used to apply VI in prob. models assuming a flexible distribution represented in terms of particles. The paper describes how this method has been implemented in NumPyro, a software for prob. programming. The method is evaluated and compare to other alternative prob. languages such as PyMC3 that uses also methods based on Stein VI.


**Summary Of The Review:**

A nice implementation of an already known method that could be used for machine learners using prob. programs.

---

> ### Author Response · Authors · 2021-11-16
> **Authors' response**
>
> We give detailed responses below. In short, ELBO-within-Stein is a novel algorithm, the program transformations that utilize Jax for Stein mixtures are novel, and the application of Stein-mixtures to DMMs is also novel. We acknowledge that we have inadvertently presented ELBO-within-Stein as the work of Nalisnick and Smythe in our introduction. This is not the case. While SVGD and EinSteinVI are closely related algorithms, amortized SVGD and OPVI are very different approaches (despite both using Stein’s method). After closer investigation and contrary to what we claimed in the paper, it turns out that there are no algorithms based on Stein’s method available in Edward. We have contacted the Edward developers to confirm this. We will revise the paper to better communicate the distinction between prior Stein-mixture algorithms and ours, highlight the novelty with relation to prior work and extend our BNN example with the below-mentioned systems to reflect algorithms based on Stein’s method currently available in PPLs. Note that the only PPL that falls into this category is PyMC3.
>
> > “It is not clear what is the difference between EinSteinVI and …”
> - We will revise the paper to highlight the difference between the Stein-based methods in the related work section as summarized above. For convenience, we have added a summary below as well.
>   - __Stein Variational Gradient Descent__ (SVGD) [in PyMC3]:
> The algorithm transports a set of fixed particles to a target distribution by iteratively moving them in the Stein variational gradient direction. EinStein VI works with a Stein mixture, ie. the particle's parameterized guide. These guides approximate the target distribution. Note that we can perform SVGD in EinSteinVI by using a point mass distribution as our guide.
>   - __Amortized Stein Variational Gradient Descent__ [in PyMC3]:
> The algorithm trains a stochastic network to draw samples from a target distribution. The network is iteratively adjusted so that the output changes in the direction of the Stein Variational gradient (the same as SVGD). In comparison, EinStein VI transports a fixed set of particles (parameterized a variational distribution, aka guide) to the target distribution. The particles change in the Stein Variational gradient direction. Note that Amortized SVGD has not been extended to work with arbitrary guides, which is why it’s not included in EinSteinVI. This would be an interesting extension to investigate.
>   - __Operator Variational Inference__ (OPVI) [in PyMC3]:
> The algorithm optimizes operator objectives, which take functions of functions to a non-negative number. They suggest a variational objective based on the Langevin-Stein Operator, which is also used for Stein discrepancy (underlying SVGD). Unlike EinSteinVI and SVGD, OPVI is not a particle-based method. However, OPVI works with the same operator (the Langevin-Stein Operator) as SVGD, which is why we included it.
> - To summarize the difference in features between EinSteinVI and PyMC3:
> | Only EinSteinVI| Both EinSteinVI and PyMC3 | Only PyMC3|
> |------------------------------------------------------------------------------------|---------------------------|----------------------------------------------------------------|
> | Stein-mixtures (Note: works with arbitrary guides, including SVGD as special case) | SVGD                      | Amortized SVGD (Note: does not work with guides)               |
> | ELBO-within-Stein                                                                  | RBF Kernel                | OPVI (Note: uses Stein’s method, but is actually not Stein VI) |
> | Kernels (matrix, vector, scaler) |  | |
> | Non-linear Stein | |   |
> | Higher order optimization|                           |                                                                |
>
> > “, it does not compare results with Edward. “
> - We mistakenly thought OPVI in Edward was publicly available in a separate project. However, this appears not to be the case. We have contacted the Edward developers to confirm this.
> - OPVI ( PyMC3) is quite different from SVGD and Stein mixtures. Nonetheless, we will extend the results for BNNs with all PPL algorithms based on Stein's divergence to give a fuller picture. _That is, we will revise the submission with results from SVGD (PyMC3), amortized SVGD (PyMC3).~, and OPVI (PyMC3).~
>
> > “The novelty of the paper is low since it does not describe any new method…”
> - The concern about novelty was brought up by several reviewers. We acknowledge that we have inadvertently presented ELBO-within-Stein as the work of Nalisnick and Smythe in our introduction. This is not the case. We highlight the substantial novelty of this work in the response to the first reviewer (Hnfk). _We will correct this error in our revision and highlight the novelty of our work._

---

> > ### Comment · Reviewer_KQPj · 2021-11-24
> > **Response to Rebuttal**
> >
> > I would like to thank the authors for the response provided. However, the now claimed to be new method seems to be based on a small modification of the approach described by Nalisnick and Smythe. This modification is not motivated in any way and the reader cannot understand why the ELBO is a good replacement for the weighted average considered originally. The authors also do not describe what is the expression for the ELBO. The proposed modification seems to be ad-hoc. The authors have to better explain what is the reason for that modification and why it is expected to perform better. My overall impression is that this paper is still at an early stage and needs more work on it.

---

> > > ### Author Response · Authors · 2021-11-30
> > > **Authors' response**
> > >
> > >  > “The proposed modification seems to be ad-hoc.”
> > > - ELBO-within-Stein follows from lower bounding $\log E[\frac{p(X,\theta)}{q(\theta|z_j)}] \geq E[\log\frac{p(X,\theta)}{q(\theta|z_j)}]$ using Jensen's inequality. Hence, the modification is principled and well-justified rather than “ad hoc”.
> > >
> > > > “The authors have to elaborate on the reasons for that modification and why it should improve performance.”
> > > - Using the ELBO is less computationally and memory intensive than importance-weighted gradient estimation because it avoids computing the importance weights and only requires one gradient computation.
> > > - While importance weighted gradient estimation provides a tighter lower bound of $\nabla_{z_j} p(x|z_j)$ than ELBO as [Burda et al.](https://arxiv.org/pdf/1509.00519.pdf) shows. [Rainforth et al.](https://arxiv.org/pdf/1802.04537.pdf) demonstrate that increasing the number of samples degrades the signal-to-noise ratio, which inevitably deteriorates the overall learning process by increasing the relative variance of the actual gradient. See [Rainforth et al.](https://arxiv.org/pdf/1802.04537.pdf), Figure 2 and pages 5.
> > > - In conclusion, the two reasons for introducing ELBO-with-Stein are (a) increased computational efficacy and (b) decreased variance of the gradient estimates.

---

> > > > ### Comment · Reviewer_KQPj · 2021-11-30
> > > > **Response to Authors**
> > > >
> > > > I would like to thank the authors for their response. I now understand the motivation for the proposed approach. However, it will be better if the paper included experiments to verify it. In particular, the decreased variance of the gradient estimates.

---

### Official Review · Reviewer_uwM1 · 2021-11-01

**Correctness:** 4
**Technical Novelty And Significance:** 2
**Empirical Novelty And Significance:** 2
**Recommendation:** 3
**Confidence:** 4

**Main Review:**

Although I am fully supportive of having source code presented in academia, I do not think this paper brings anything of particular interest, in particular for this venue.
Although the introduction to the different Stein-VI methods is nicely written and very understandable, this paper does not aim to be a review which is already done in Anastasiou 21' as the authors point out.
What the paper could bring, would be novel techniques or implementations to improve speed, but unfortunately only using `vmap` to solve things seems a bit light.
Another major weakness is that no code is available, for a submission on a software this is unacceptable.

The positive point I found was the generalization of these different algorithms through a unified algorithm, but this alone is not enough to make the work sufficiently significant to be accepted.


**Summary Of The Paper:**

The paper presents the implementation of a framework to work with different "Stein VI" methods in NumPyro.
A quick introduction to the different algorithms is given as well as an evaluation of a series of examples.

**Summary Of The Review:**

The paper does not bring anything interesting enough research-wise and such a paper should be submitted to a software venue/journal like [JOSS](https://joss.theoj.org/).

---

> ### Author Response · Authors · 2021-11-16
> **Authors' response**
>
> We give detailed responses below. In short, ELBO-within-Stein is a novel algorithm, the program transformations that utilize Jax for Stein mixtures are novel, and the application of Stein-mixtures to DMMs is novel. We acknowledge that we have inadvertently presented ELBO-within-Stein as the work of Nalisnick and Smythe in our introduction. This is not the case. The (anonymised) source code is available here. _We will add more detail to the core algorithm section in our revision._
>
> > “I do not think this paper brings anything of particular interest”
> - The concern about novelty was brought up by several reviewers. We highlight the substantial novelty of our work in the response to the first reviewer (Hnfk).
>
> > “implementations to improve speed”
> - EinSteinVI is significantly faster than any available SVGD implementation. See BNN example and reported times in SVGD.
>
> > “unfortunately only using vmap to solve things seems a bit light”
> - We seem to have oversimplified the description of EinSteinVI in the article for the sake of clarity: the implementation of EinsteinVI required novel approaches beyond the simple application of vmap. As can be [seen](https://github.com/einsteinvi/einsteinvi_iclr/blob/main/numpyro/contrib/einstein/stein.py#L168-L281) (https://github.com/einsteinvi/einsteinvi_iclr/blob/main/numpyro/contrib/einstein/stein.py#L168-L281) by inspecting the source code, the complexity in EinSteinVI is in the transformations being mapped. This is detailed in the response to the first reviewer (Hnfk). We will add more detail to the core algorithm section in our revision.
>
> > “Another major weakness is that no code is available”
> - The source code is already publicly available. We've made an anonymous GitHub repo with EinSteinVI [here](https://github.com/einsteinvi/einsteinvi_iclr) (https://github.com/einsteinvi/einsteinvi_iclr).

---

### Official Review · Reviewer_N78P · 2021-11-02

**Correctness:** 3
**Technical Novelty And Significance:** 2
**Empirical Novelty And Significance:** 2
**Recommendation:** 5
**Confidence:** 3

**Main Review:**

# Strength
- Integration of SteinVI into numpyro seems very useful. Users can easily take advantage of the state-of-the-art SteinVI algorithms for their own Bayesian modelings. Moreover, experimental results show that it is faster than the existing implementation.
- The concept of ELBO-within-Stein is innovative and seems interesting to be explored.
- Extending the stein mixture method to deep Markov models is a novel application.

# Weakness
- No code example is presented. Thus I could not tell how EinSteinVI is easier or better than the standard Numpyro and PyMC3.
- Related to the above, I was not convinced by the motivation of this work in the Introduction. The authors explained that
> Currently Stein VI methods are available in PyMC3 (Salvatier et al., 2016) and Edward (Tran et al., 2016), neither of which include the advanced methods available in EinSteinVI. $\ldots$ While algorithmic power is growing, a distinct lack of integration with a general probabilistic programming language (PPL) framework remains. Such integration would solve one of the most prominent limitations of traditional VI: its lack of flexibility in capturing rich correlations in the approximated posterior

-  As far as I understood, these phrases correspond to the motivation of this work. But after reading this paper, I thought that why PyMC3 or naive Jax implementation is not enough to resolve the above issues and why proposed EinSteinVI resolves them differently from PyMC3 or Jax implementation. Add an explanation or show additional experimental results for that.

- Some important Stein VI methods seem lacking, e.g,
    - Function space particle optimization for bayesian neural network (https://arxiv.org/abs/1902.09754)
    - Understanding and Accelerating Particle-Based Variational Inference (https://arxiv.org/abs/1807.01750)
    - Learning Equivariant Energy Based Models with Equivariant Stein Variational Gradient Descent (https://arxiv.org/abs/2106.07832)
- The presentation of the paper is poor. I think the concept of ELBO-within-Stein and its application to deep Markov models are the technical contributions of this work, but they are poorly explained.
- No experiments to support the usefulness of EinSteinVI for Non-linear Stein VI, Matrix-valued kernel stein VI, and message passing stein VI.


# The reason for the score
Although the developed library seems useful for the community and the concept of ELBO-within-Stein seems interesting, I think the contribution is not enough experimentally and theoretically as I described in Weakness above.


# Comments
- The bandwidth tuning is critical for Stein VI. I recommend adding the tuning method other than the median method, e.g, the heat kernel approach introduced in Understanding and Accelerating Particle-Based Variational Inference (https://arxiv.org/abs/1807.01750)
- I think the presentation can be improved to make this work more attractive.  The technical contribution, that is, the concept of ELBO-within-Stein and its application to deep Markov models, should be more emphasized. The authors should explain how novel they are in more detail. Also, the authors should add more experiments to support the usefulness of the concept of ELBO-within-Stein and stein mixtures in deep Markov models.

**Summary Of The Paper:**

This paper proposed a new probabilistic programming framework for stein variational gradient descent and its variants.  The framework put its basis on Numpyro. Experimental results show that this framework shows faster computation for inference in wall clock time. Moreover, using this framework, the authors developed a new Stein mixture algorithm for deep Markov models, which shows better performance than existing methods.


**Summary Of The Review:**

The contribution is not enough experimentally and theoretically to be published. Although the proposed framework seems beneficial for the community, it lacks a detailed comparison regarding the usability or performance compared to existing frameworks, e.g., PyMC3. Regarding the technical contribution, the extension of stein mixtures seems promising, but it lacks the explanation, numerical experiments, and theoretical justification.

---

> ### Author Response · Authors · 2021-11-16
> **Authors' response**
>
> Below we give a detailed response to points raised by the reviewer. To summarize, unlike currently available tools for SteinVI, EinStein works with guide programs. We plan to maintain and extend EinSteinVI, but point out that EinSteinVI in its current form presents novel research in addition to offering a useful tool for practitioners. We acknowledge that we have inadvertently presented ELBO-within-Stein as the work of Nalisnick and Smythe in our introduction. This is not the case.  _We will revise our submission by highlighting our contributions, adding a code example, including Non-linear Stein VI in the BNN example, and elaborating on our DMM example._
>
> > “No code example is presented”
> - In response to the first reviewer (Hnfk) we show how similar the EinSteinVI interface is to SVI. _We will revise the paper with a small code example._
>
> > “The presentation of the paper is poor... “
> - We acknowledge we have inadvertently attributed the ELBO-within-Stein to Nalisnick and Smyth and thank the reviewer for picking up on this. We detail the difference between ELBO-within-Stein and Nalisnick and Smyth’s method in our response to the first reviewer (Hnfk). _We will of course correct this error in our revision and highlight the novel contributions that are ours._
>
> > “why proposed EinSteinVI resolves them differently from PyMC3 or Jax implementation.”
>
> - The core difference is that EinSteinVI works with arbitrary guide programs. No current PPL supports for Stein VI with guides. Experimental evidence shows that Stein mixtures provide a more expressive class of models than SVGD. With EinSteinVI we can efficiently infer them.
>
> > “No experiments to support the usefulness of EinSteinVI for Non-linear Stein VI…”
> - The star distribution achieves the lowest MMD with the matrix RBF kernel preconditioned with the Hessian matrix. This shows the advantage of a matrix-valued kernel in Stein VI.
> - It's correct that we do not have examples for message passing Stein VI and Non-linear Stein VI. _We will revise our submission with Non-linear Stein VI included in the BNN example._
>
> > “ Some important Stein VI methods seem lacking, e.g,"
>
> - Thanks for bringing these papers to our attention. We will look into integrating them in EinSteinVI. We believe EinSteinVI provides the ideal platform for implementing these and future algorithms related to Stein VI. We intend to continually extend EinSteinVI through our research and by integrating new methods from the community. We plan that EinSteinVI will become a unified codebase for researchers and an efficient black-box inference engine for practitioners.
>
> > “The bandwidth tuning is critical for Stein VI”
>
> - We agree on the importance of bandwidth tuning for the RBF kernel and we will definitely add more advanced tuning methods to the backlog. The kernel interface is modular to inference so kernels can be changed as long as they keep their signature. Suggestions such as these are exactly why we believe a project such as EinSteinVI is warranted.
>
> > “The technical contribution, that is, the concept of ELBO-within-Stein and its application to deep Markov models should be more emphasized.“
> - _We acknowledge that the DMM example should be elaborated further and will do so in our revision._

---

### Official Review · Reviewer_Hnfk · 2021-11-04

**Correctness:** 3
**Technical Novelty And Significance:** 2
**Empirical Novelty And Significance:** 2
**Recommendation:** 3
**Confidence:** 4

**Main Review:**

[Strength]

The paper is overall well-written and the method is clearly explained. The literature review is thorough.


[Weakness]
- Motivation is unclear to me. Why implementing the Stein Variational inference methods in PPL framework is preferable to non-PPL framework? And also why NumPyro is preferable to other PPLs? In the introduction, it is briefly mentioned that "Such integration would solve one of the most prominent limitations of traditional VI: its lack of flexibility in capturing rich correlations in the approximated posterior", which I consider as one of the motivations for using PPL framework. Still, theoretical or empirical evidence for this statement is missing and this work would be more appealing if the authors elaborate on it.

- The originality is low since all the Stein variational inference algorithms implemented in NumPyro in this work are proposed by prior work and the authors do not propose any new algorithm.

**Summary Of The Paper:**

The aim of this work is to integrate the Stein variational inference methods in the existing probabilistic programming language NumPyro. The implemented methods include variants of Stein variational gradient descent algorithms with a range of kernel functions, non-linear scaling of update terms, and matrix-valued kernels. Empirical results with existing baselines for real-world problems are provided.


**Summary Of The Review:**

My main concern is that the original contribution of this work is not significant enough for being published as a conference paper.

---

> ### Author Response · Authors · 2021-11-16
> **Authors' response**
>
> Below we give a detailed response to points raised by the reviewer. We point out that the ELBO-within-Stein algorithm, several methods used for its implementation, and the Stein mixture DMM are each novel contributions. We acknowledge that we have inadvertently presented ELBO-within-Stein as the work of Nalisnick and Smythe in our introduction. This is not the case: ELBO-within-Stein is actually entirely new and not previously described. We chose NumPyro because (a) its SVI API provided a suitable template for EinSteinVI and (b)   Numpyro’s computational backend is the high-performance framework Jax, which is required for the efficient implementation of transformations between the Jax array data structure and NumPyros dictionary data structure. _We will revise our submission with a justification for choosing NumPyro, a code example, and elaborate our DMM example._
>
> > “Motivation is unclear to me…”
>
> - We implemented EinSteinVI in NumPyro for the following reasons:
>   - Python is the de facto language for data science.
>   - NumPyro is a PPL that provides the data structures required to track and manipulate probabilistic programs.
>   - NumPyro allows the execution of arbitrary Python code in models and guides.
>   - NumPyro uses Jax as its computational backend. To our knowledge, Jax is currently the fastest deep learning framework in python.
>   - Stein mixtures allow parameterization of variational distributions using particles. Particle methods can capture rich correlations as they will move according to the probability surface.
> EinStein’s API can be kept very close to the SVI API of NumPyro. Using Stein mixtures instead of SVI requires changing only one line and choosing a kernel. Below is a concrete example:
> ``` python
> inf = SVI(model, guide, Adam(lr=.1),  trace_ELBO())
> inf.run(rng_key, num_iter, model_args)
> ```
> becomes
> ``` python
>      inf = Stein(model, guide, trace_ELBO(), Adam(lr=.1), RBFKernel())
>      inf.run(rng_key, num_iter, model_args)
> ```
> - We will revise the paper with a clear justification for NumPyro as our target and include a code example for usage.
>
>
> > “The originality is low since all the Stein variational inference…”
> - We acknowledge that we have inadvertently presented ELBO-within-Stein as the work of Nalisnick and Smythe in our introduction. This is not the case.
>  - The novelty in the paper is three-fold:
>    1. Nalisnick and Smyth approximates the stein force by $$E_{z_j \sim q_{Z}(z)}[k(z_i, z_j) \sum_{s=1}^S w_s \frac{d}{d z_j} \log (\frac{p(X,\theta_s)}{q(\theta_s|z_j)})] + S_{Z}^{-}(z_i),$$ ELBO-within-Stein approximates the Stein force by $$ E_{z_j \sim q_{Z}(z)}[k(z_i, z_j) \frac{d}{dz_j} \mathcal{L}(z_j)] + S_{Z}^{-}(z_{i})$$ where $\mathcal{L}$ is the ELBO. Our formulation uses a single sample and avoids computing the importance weight tilde{w}_s. Tilde{w}_s is computed from the marginal likelihood by drawing from a differentiable non-centered parametrization of the variational posterior which is much more complex. We show experimentally that its performance is on par with Nalisnick and Smyth’s method on the BNN example.
>    2. The transformation between NumPyro's data structure for tracking variables and a differentiable program in Jax is novel. A naive transformation will incur a massive memory overhead when computing the Jacobian for transforming between the unconstrained and constrained space. We have resolved this problem by appropriately decomposing the monolithic particle (the tensor representation in Jax) according to the sample sites. Our transformation allows EinSteinVI to benefit from Jax while maintaining NumPyro’s flexibility in expressing models (and guides).
>    3. The application of Stein mixtures to deep Markov models is novel and its performance is better than the current state-of-the-art method by Jankowiak and Karaletsos (2019), which is based on Adaptive Velocity Fields gradient estimator.
> - _We acknowledge the DMM example should be elaborated and will do so in our revision._

---

> > ### Comment · Reviewer_Hnfk · 2021-12-01
> > **Acknowledgment**
> >
> > I truly appreciate the response from the authors, which helps make clear why they chose NumPyro and that ELBO-within-Stein is actually one of the contributions in this work. Still, I echo with the other reviewers that more efforts are required to illustrate why ELBO-within-Stein is preferred over the existing work. This would lead to non-trivial modification to this work and thus I feel that it is not yet ready for being published.

---

### Author Response · Authors · 2021-11-22
**Changelog**

## Rewrites
1. Made ELBO-within-Stein into stein-mixtures section (describing nalisnick and smyth) and ELBO-wihtin-Stein (ours).
2. Added description of Stein mixtures to Stein-mixture section
3. Removed inadvertent reference to Nalisnick and Smyth in contribution summary
4. Add descriptions of other methods to related work
5. Moved Related works after Stein-mixtures
6. Added code example
7. Rewritten introduction to highligh our contributions
8. Added description of constaint <-> unconstraint jacobian in Implementation

## Errata
1. changed S_Z^+ gradient index to j (two force of SVGD).
2. changed S_Z^+ gradient index to j (matrix valued-kernels).
3. Removed reference to OPVI in Edward.

## Benchmarking
1. Tuned temperature for SVGD in PyMC3
2. Added Amortized SVGD
3. Updated DMM to latest version of EinSteinVI

---

### Decision · Program_Chairs · 2022-01-20

**Decision:**

Reject

**Comment:**

The paper aims to integrate Stein variational inference methods into the existing probabilistic programming language NumPyro. The implemented methods include variantions of Stein variational gradient descent with different types of kernel functions, non-linear scaling of update terms, and matrix-valued kernels. The paper includes empirical results with a comparsion with existing baselines in real-world problems. Using this framework, the authors developed a new Stein mixture algorithm for deep Markov models, which shows better performance than existing methods.

Strengths:

- The paper is overall well-written and the method is clearly explained.
- The literature review is thorough.
- Integration of SteinVI into numpyro seems useful. Users can easily take advantage of the state-of-the-art SteinVI algorithms for their own Bayesian modelings.
- Extending the stein mixture method to deep Markov models is a novel application.

Weaknesses:

- The originality is low the authors propose algorithms that are very similar to previous work and there is a lack of experiments to verify the usefulness of the proposed method, for example,
  to verify the decreased variance of the gradient estimates claimed by the authors.
- Efforts are required to illustrate why ELBO-within-Stein is preferred over the existing work.
- Some important Stein VI methods seem lacking.
- No experiments to support the usefulness of EinSteinVI for Non-linear Stein VI, Matrix-valued kernel stein VI, and message passing stein VI.

All reviewers vote for rejection. I recommend the authors to addrss the limitatoins mentioned above and improve the paper before its resubmission to another venue.